# On the Primary Water Radicals’ Production in the Presence of Gold Nanoparticles: Electron Pulse Radiolysis Study

**DOI:** 10.3390/nano10122478

**Published:** 2020-12-10

**Authors:** Viacheslav Shcherbakov, Sergey A. Denisov, Mehran Mostafavi

**Affiliations:** Institute de Chimie Physique (ICP), CNRS/Université Paris-Saclay, Bât. 349, 91405 Orsay, France; viacheslav.shcherbakov@universite-paris-saclay.fr

**Keywords:** radiosensitization, radicals, gold nanoparticles, radiolysis

## Abstract

Gold nanoparticles are known to cause a radiosensitizing effect, which is a promising way to improve radiation therapy. However, the radiosensitization mechanism is not yet fully understood. It is currently assumed that gold nanoparticles can influence various physical, chemical, and biological processes. Pulse radiolysis is a powerful tool that can examine one of the proposed effects of gold nanoparticles, such as increased free radical production. In this work, we shed light on the consequence of ionizing radiation interaction with gold nanoparticles by direct measurements of solvated electrons using the pulse radiolysis technique. We found that at a therapeutically relevant gold concentration (<3 mM atomic gold, <600 μg × cm^−3^), the presence of gold nanoparticles in solution does not induce higher primary radicals’ formation. This result contradicts some hypotheses about free radical formation in the presence of gold nanoparticles under ionizing radiation previously reported in the literature.

## 1. Introduction

Ionizing radiation is part of numerous health applications in diagnostics and therapeutics. Radio treatment (RT) is used in an overwhelming number of cases along with other cancer-fighting methods: surgery, chemotherapy, and immunotherapy. RT’s main development challenge is to reduce side effects, which is addressed by better dose localization and attempts to decrease the required dose by utilizing special radiosensitizing agents. The radiosensitizers could be drugs, macromolecules, oxygen and its mimics, or high Z-materials, including gold nanoparticles (AuNPs) [1]. AuNPs are desirable material for bioapplications due to their facile synthesis, small size, versatile surface chemistry, easy cell membrane penetration properties, and low toxicity [2].

First experiments on gold as a dose enhancer were done using foils [3]. This study was focusing on the interface effects occurring between high-Z material and cell monolayer under X-ray radiation. The next step was to utilize higher surface area materials such as microparticles. Gold microspheres (1.5–3 µm, up to 3 wt %) were used for curing tumors in mice. The radiosensitizing effect was attributed to a physical dose enhancement, which was verified by Fricke dosimetry. The dose enhancement effect was present only for X-ray ionizing radiation, but not for γ-rays (^137^Cs) [4]. Later the nanoparticles of gold were used for radiosensitization studies [5]. This seminal work provoked a cascade of studies on the radiosensitization effect using nanomaterials. The interest in this subject from the medical, radiation chemistry, and biology communities was growing since then for decades [6,7,8,9].

As discussed elsewhere [8,10], the initially used hypothesis on dose enhancement due to higher cross-sections of high-Z materials towards ionizing radiation (X-rays range) compared to low-Z elements was unable to explain observed results. The problem is that the concentration of AuNPs in vivo and in vitro experiments is less than required to deposit a significant amount of energy into gold particles. In the early works [4,5], a much higher concentration (0.3–3 wt %) of gold was used with a combination of X-rays. In more recent works, particles’ delivery is targeted within the cell interior with much lower concentrations (ca. 100 µg/mL, 0.01 wt %), utilizing different radiation sources such as γ-rays and protons, not only X-rays [9]. However, the radiosensitizing effects, sometimes higher than 50% [9], were observed in the presence of AuNPs even at a low gold concentration and ionizing radiation energy well above the X-ray range, where the cross-section of gold and water or soft-tissue are alike [11]. The fact that the observed results did not correlate with the type of ionizing radiation and its energy urged scientists to look for other explanations. To date, the biological effects of gold nanoparticles are actively studied [7,8,9,12]. One of the observed effects in the presence of AuNPs under ionizing radiation is higher oxidative stress, which causes cell biochemistry disruption and death. These radiobiological effects are not easy to unravel due to cellular chemistry’s complexity, including signal pathways responsible for the bystander effect [13,14]. The oxidative stress is often understood in the context of higher dose absorption, which causes a higher concentration of reactive species, mainly oxygen-containing (ROS), and the main focus is devoted to hydroxyl radical (^•^OH) [6,9].

Interestingly, several experiments with molecular systems support the idea of increased free radicals production in the presence of AuNPs. The additional radical production was always higher than predicted by physical dose enhancement and was observed for different radiation types and a broad range of energies [6,15,16]. Therefore, other explanations were proposed, such as special properties of interfacial water leading to higher radiolytic yields of primary radicals; a higher concentration of oxygen-rich species on the surface of AuNPs, leading to higher ROS production [6].

Pulse radiolysis is a suitable technique for the direct detection of free radicals’ overproduction in homogenous solutions. Therefore, in this work, we focus on the radiolysis of water in the presence of AuNPs. The firm conviction dictates the necessity of such work that the radiosensitization mechanism explanation complexity must be reduced. The development of pulse radiolysis techniques allows us to directly determine the radiolytic yield of hydrated electrons (e^−^_aq_) precisely and indirectly that of ^•^OH radicals, just a few picoseconds after the energy deposition in water. Herein, we report the absence of AuNPs’ effect on primary water radicals formation and discuss the value of these results on the explanation of the AuNPs’ radiosensitizing effect.

## 2. Materials and Methods

### 2.1. Electron Pulse Radiolysis

The photocathode-driven electron accelerator ELYSE (University of Paris-Saclay) operating at 5 Hz delivering 5 ps electron pulses (45–100 Gy) at 7 MeV was used in this study. The deposited dose per pulse was deduced from e^−^_aq_ absorbance measurements in pure water and verified before each set of scans. More details on the experimental setup could be read elsewhere [17,18]. The stroboscopic detection with a mechanical delay line was used for time-resolved studies in the range of 10 ns. For longer timescales, a transient absorption setup with a streak camera detection was used. For all experiments, the optical quartz cells of 5 mm length were used with thin (100 μm) optical windows.

Optical detection during pulse radiolysis measurements of AuNP suspensions has some limitations. The gold nanoparticles suspensions strongly absorb in the UV–vis range. Thus, only the red light >600 nm can penetrate the samples. The solutions of 30 mL were circulated using a peristaltic pump with a flow rate of 40 mL × min^−1^ for measurements in microsecond timescales, whereas, for subnanosecond scales, static quartz cells were used.

Utilizing pulse radiolysis coupled with time-resolved spectroscopy for AuNPs suspensions studies requires the correct reference, where pure water may not be a suitable one. The time evolution of radical species in AuNPs suspensions can be affected by the presence of nanoparticle stabilizers. Thus, a more relevant reference is a supernatant obtained by centrifugation of AuNPs suspension, eliminating gold nanoparticles to a large degree but leaving other chemical substances. We controlled by absorption measurement that at least 99% of AuNPs were removed from the supernatant solution (Appendix A). Although for picosecond and nanosecond timescales, where bimolecular reactions are not significant, pure water could be appropriate as a reference. Our experimental setup allows easily for the determination of optical density within 1 mO.D. accuracy for a nanosecond and shorter timescales, and 5 mO.D. for a microsecond one. Thus, the hydrated electron concentration could be determined with an accuracy of 50 nM and 250 nM, respectively.

### 2.2. Gold Nanoparticles Synthesis

Gold nanoparticles were prepared by two widely known methods: reduction by sodium borohydride [19] and sodium citrate (Turkevich method) [20]. The final gold concentration was 3 mM (atomic concentration) or 600 µg/mL in both cases. Reduction by sodium borohydride included the following steps: 3 mL of 100 mM solution of K[AuCl_4_] was mixed with 87.7 mL of deionized water and 9.3 mL of 100 mM solution of NaBH_4_ was added to the solution under stirring by magnet stir bar with high speed. After the reduction of gold ions, the solution changed color from yellow to dark red. After the borohydride was added, stirring was continued for 5 min. For the Turkevich method, 3 mL of 100 mM solution of K[AuCl_4_] was mixed with 82.5 mL of deionized water. The solution was heated up to 100 °C, then 4.5 mL of 100 mM of sodium citrate was added to the solution under stirring by magnet stir bar with high speed. When the solution has turned red, heating was stopped. The solution was cooled down under stirring. The full reduction of gold ions was verified by the total disappearance of the absorption at 220 nm. All chemicals were purchased from Sigma-Aldrich (St. Quentin Fallavier, France).

The solution of AuNPs prepared by borohydride reduction method had pH 8. The solution of AuNPs prepared by citrate reduction method had pH 4, which was adjusted by NaOH to 5.4. The UV–vis absorption spectra of AuNPs, their size distribution, zeta-potentials, and TEM image are presented in the Appendix A.

## 3. Results and Discussion

In the present work, two types of AuNPs were used prepared using borohydride and sodium citrate (Turkevich) as reduction agents. Both techniques allowed us to synthesis stable AuNPs with an atomic concentration of gold up to 3 mM (600 µg/mL) and with simplistic small molecule/ion stabilization. The size of particles synthesized by the borohydride reduction method was 20 nm (Appendix A, the absorption maximum was 520 nm and the zeta potential was −20 mV). Those particles prepared by the Turkevich method had a mean hydrodynamic diameter of 45 nm (Appendix A, the absorption maximum was 533 nm and the zeta potential was −35 mV).

The radiolysis of water after energy deposition could undergo two pathways (Scheme 1). The main path was ionization, and the minor one was a dissociation of an excited water molecule. After ionization, the survived water-cation radical loses a proton, leaving behind ^•^OH radical [21]. A secondary electron thermalizes within a few femtoseconds, followed by its solvation within 1 ps. Dissociation of the excited water molecule leads to the formation of the ^•^H atom and ^•^OH radical. The formed species inside ionization spurs react with each other and solutes present in the solution. Besides, one must always remember that spurs tend to increase in volume due to radicals’ diffusion, which decreases the possibility of such reactions.

Direct detection is possible for the hydrated electron, which has a high molar absorption coefficient [23]. We found no difference in hydrated electrons production through picosecond pulse radiolysis in water and the solution of AuNPs (with the highest possible concentration for the measurements, 3 mM) on a time scale of a few tens of picoseconds, where radical–radical reactions did not occur (Figure 1 and Appendix A). The hydrated electron concentration just after the 5 ps electron pulse correlates with 90% of ^•^OH radicals (ionization path, a yield of the hydrated electron at 10 ps was 0.44 μmol/J [24] and ^•^OH 0.52 μmol/J [21,24]).

One could claim that in the presence of AuNPs, the water dissociation channels could be altered due to the unique properties of interfacial water molecules, which could lead to higher ^•^OH and ^•^H production. This hypothesis came to be invalid because the decays of hydrated electrons on nanosecond and microsecond scales were not affected (Figure 2 and Figure 3 and Appendix A). This observation indicates that the radical–radical reactions occurred in the radiation track in the same way as in the solution without AuNPs (Scheme 1).

We selected supernatants as references for a microsecond range because they contained most of the solutes, which could react with e^−^_s_ and/or ^•^OH radicals. In the case of AuNPs prepared by the borohydride reduction method, the decay of hydrated electrons was slower in AuNPs suspension and its supernatant than in neat water with the same pH (Figure 3A). It is explained by the fact that anions BO_3_^3−^ and BH_4_^−^ react with the ^•^OH radical [25], suppressing the ^•^OH radical-hydrated electron interaction, which increases the lifetime of e^−^_aq_. In contrast to the AuNPs solution prepared by the Turkevich method, the decay in the supernatant and AuNPs suspension was faster than in water (Figure 3B) because hydrated electrons reacted with acetonedicarboxylic acid (DCA) generated during the synthesis as a byproduct of citrate oxidation [26]. Absorption at 260 nm in the supernatant verified the presence of DCA (Appendix A). The rate constant of the hydrated electron reaction with DCA is expected to be similar to the acetone reaction (6.5 × 10^9^ M^−1^s^−1^) [25]. This observation demonstrates the absence of any primary radical overproduction reported for similar AuNPs by indirect measurements [6,15] or presolvated electron scavenging reported for a higher concentration of AuNPs stabilized by ionic liquids, by both muon methods and pulse radiolysis [27,28]. Previously the absence of ^•^H overproduction in the AuNPs solutions stabilized by ionic liquids was reported [28].

Similar experimental results, namely the absence of increased primary radicals’ production studied by pulse-radiolysis, were obtained for silver nanoparticles (AgNPs). The experimental data, synthesis, and characterization methods for those samples are presented in the Supporting Materials (Appendix A).

The conducted experiments were different from those in vitro and in vivo due to extreme dose rates, exceeding 10^13^ Gy/s. At such high dose rates, the lifetime of radicals was significantly reduced, e.g., the hydrated electrons were entirely disappearing within ca. 1 µs (Figure 3) due to a high concentration of later and ^•^OH radicals in the order of micromoles per liter. The concentration of AuNPs lay in the range of 12 nM, and thus the interaction between radicals and particles, which is discussed elsewhere [29,30], was a statistically rare event in our conditions. Therefore, observed decays of hydrated electrons in the microsecond range were explained by radical chemistry mainly within the radicals’ spurs and diffusion into the volume of the solution. If AuNPs cause higher dose absorption, then it must lead to a higher primary radicals’ formation, which could eventually result in a higher concentration of ROS later in time. When ionizing radiation is applied, the higher production of secondary electrons occurs around AuNP compared to water. For X-ray radiation, the dose absorption is two orders of magnitude higher due to the higher cross-section for gold compared to water molecules. For γ quants and high-energy particles, the increase of secondary electrons production occurs because those low energy electrons emitted from AuNP have higher LET values compared to those generated in water. Thus, more primary radicals are produced around single nanoparticles, no matter what kind of radiation is applied. To have a 10% difference in the absorbed dose between water and AuNPs suspension at X-ray energies where the cross-section of gold is the highest (35 keV) [31], the concentration of AuNPs must be in the order of 3 mM of the atomic gold concentration. At the same concentration of gold under higher energy ionizing radiation, the dose absorbance enhancement will be in the order of 0.06%. It is almost impossible to detect such a difference with direct or even more difficult for indirect measurements.

Of course, such a way of thinking is applicable only for homogenous solutions, and it is different for particles’ aggregates usually formed in the cell. In many works, fluorescent dyes were used to detect ROS [6]. Some of these works report a 4.5-fold enhancement of ^•^OH radical production even for 1.25 MeV gamma rays, which have a similar linear energy transfer as 7 MeV electrons used in our work [15]. In the fluorescent method, the sacrificial molecule is converted to a fluorescent molecule reacting with ROS, such as the ^•^OH radical [32]. Our results contradicted the proposed explanations. Only the fluorescent molecule formation was measured in the works utilizing fluorescent dyes, while the initial molecule’s disappearance was missed [6]. In our understanding, the authors misinterpret their experimental results since they did not consider gold nanoparticles’ catalytic properties towards organic radicals formed under ionizing radiation [33,34,35], which could affect the fluorescent product’s mechanism formation.

We would like to stress out that the results of this paper must be utilized with care for cellular systems, where local dose enhancement around the AuNP must be taken into account for the radiosensitization effect.

## 4. Conclusions

To conclude, it was shown that in the presence of 20 nm and 45 nm AuNPs and AgNPs at a therapeutical concentration (≤3 mM atomic concentration) in homogeneous solutions under 5 ps electron pulse (7 MeV), there was no detectable increase of primary radicals’ formation and as a result, another ROS production increase was not expected. In this way, to explain the effects of AuNPs under ionizing radiation such as radiosensitization requires other mechanisms than an increase of radiolysis yield of primary radicals. Such mechanisms of AuNPs’ action must occur not at the timescale of energy deposition, primary radicals’ formation, and their further evolution, but later in a homogeneous step of water radiolysis. This could be related to the chemical activity of gold nanoparticles, namely, catalysis. We will thoroughly analyze this subject in future work that is in progress.

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
