# Peer review of "On the Primary Water Radicals’ Production in the Presence of Gold Nanoparticles: Electron Pulse Radiolysis Study"

_nanomaterials, 2020, doi:10.3390/nano10122478_

Round 1

Reviewer 1 Report

This article presents a pulse radiolysis experimental investigation of water radiolysis in the presence of gold nanoparticles.  The preparation of the nanoparticles and pulse radiolysis technique are well detailed and comprehensive nanoparticle quantification and pulse radiolysis results are presented. The conclusions drawn are soundly based on the experimental results presented and the article is well written. The results presented in the article are of great value and interest, providing evidence that there is no increase in radiolysis yield in the presence of gold nanoparticles. This invalidates previous common radiosensitisation theories about increased radiolysis yield from either increased ionisation around the nanoparticles or interfacial effects and directs the search for the cause of gold nanoparticle enhancement of radiation therapy to other causes. I only have a few comments.

The finding of no enhancement in radiolysis from gold nanoparticles suggests that other mechanisms, likely due to the biological effect of the nanoparticles themselves could be responsible for the observed radiosensitisation. Turnbull et al (ACS nano 13 (5), 5077-5090 2019) observed the downregulation of processes involved in DNA damage repair from the presence of nanoparticles which may explain the radiosensitisation effect and the limited biological effect on unirradiated cells.

The experimental setup for the pulse radiolysis likely resulted in a disperse nanoparticle distribution. Often nanoparticles will accumulate within clusters when distributed within a cell. While this would not result in a increase in reactive species production on a larger scale as confirmed in the experiment it could potentially change the distribution to enable a concentration of reactive species production in close proximity to the cluster potentially enabling enhanced biological damage to biological targets in close proximity.   

Author Response

Reviewer 1

This article presents a pulse radiolysis experimental investigation of water radiolysis in the presence of gold nanoparticles.  The preparation of the nanoparticles and pulse radiolysis technique are well detailed and comprehensive nanoparticle quantification and pulse radiolysis results are presented. The conclusions drawn are soundly based on the experimental results presented and the article is well written. The results presented in the article are of great value and interest, providing evidence that there is no increase in radiolysis yield in the presence of gold nanoparticles. This invalidates previous common radiosensitisation theories about increased radiolysis yield from either increased ionisation around the nanoparticles or interfacial effects and directs the search for the cause of gold nanoparticle enhancement of radiation therapy to other causes. I only have a few comments.

The finding of no enhancement in radiolysis from gold nanoparticles suggests that other mechanisms, likely due to the biological effect of the nanoparticles themselves could be responsible for the observed radiosensitisation. Turnbull et al (ACS nano 13 (5), 5077-5090 2019) observed the downregulation of processes involved in DNA damage repair from the presence of nanoparticles which may explain the radiosensitisation effect and the limited biological effect on unirradiated cells.

The experimental setup for the pulse radiolysis likely resulted in a disperse nanoparticle distribution. Often nanoparticles will accumulate within clusters when distributed within a cell. While this would not result in a increase in reactive species production on a larger scale as confirmed in the experiment it could potentially change the distribution to enable a concentration of reactive species production in close proximity to the cluster potentially enabling enhanced biological damage to biological targets in close proximity.

Answer:

We thank the reviewer for his/her kind comments. We added several sentences in the text regarding the biological effect and mentioned that our results mainly relate to homogeneous systems. Of course, the AuNPs aggregation in the cells is out of the scope of the current work.

* The line numbers are for the mode in which all corrections (track changes) are hidden.

Line 53: To date, the biological effects of gold nanoparticles are actively studied [7–9,12].

Line 203: Of course, such a way of thinking is applicable only for homogenous solutions, and it is different for particles' aggregates usually formed in the cell.

Reviewer 2 Report

The specific study targets the radiosensitization effects of gold NPs where many different mechanisms are probably present. The authors use Pulse radiolysis where certainly a more precise approach. On the hand the limited data presented and the lack of proper Discussion and Introduction cover render this study preliminary. The ROS production is not really quantified and verified for different energy pulses, even positive controls like standard X rays of low energies. The use of different NPs sizes and PEG or citrate cover is critical. 

In addition please find other (not necessary comments):

  1. The authors need to revise the whole abstract and Introduction and provide in the abstract more specific details especially under the view of new experiments and restrain from absolute expressions throughout the manuscript like "t: Up to date, the mechanism of radiosensitization by gold nanoparticles remains unknown. The growth in publications and many suggested explanations do not clarify it". This is not true the radiosensitization mechanisms have been studied and partially trusted : secondary electron production, ROS, biological damage not only in DNA, for example membrane or mitochondria de-structure, killing via apotposis etc. These and the ideas of DDR induction and inflammatory and immune response adding to the over radiosensitization primary effects are discussed in : "Dimitriou, N.M.,  et al. (2017) Gold nanoparticles, radiations and the immune system: Current insights into the physical mechanisms and the biological interactions of this new alliance towards cancer therapy. Pharmacol Ther, 178, 1-17." . Such integrative discussions must be incorporated . 
  2. The authors need to support better the production of ROS for different NPs and energies of pulses which are currently in the MeV area. Very high. 
  3. The fact that the role of entering into the cell nucleus or as close as possible is completely disregarded opens logical questions about the use of these findings. 

Author Response

Reviewer 2

The specific study targets the radiosensitization effects of gold NPs where many different mechanisms are probably present. The authors use Pulse radiolysis where certainly a more precise approach. On the hand the limited data presented and the lack of proper Discussion and Introduction cover render this study preliminary. The ROS production is not really quantified and verified for different energy pulses, even positive controls like standard X rays of low energies. The use of different NPs sizes and PEG or citrate cover is critical. 

In addition please find other (not necessary comments):

  1. The authors need to revise the whole abstract and Introduction and provide in the abstract more specific details especially under the view of new experiments and restrain from absolute expressions throughout the manuscript like "t: Up to date, the mechanism of radiosensitization by gold nanoparticles remains unknown. The growth in publications and many suggested explanations do not clarify it". This is not true the radiosensitization mechanisms have been studied and partially trusted : secondary electron production, ROS, biological damage not only in DNA, for example membrane or mitochondria de-structure, killing via apotposis etc. These and the ideas of DDR induction and inflammatory and immune response adding to the over radiosensitization primary effects are discussed in : "Dimitriou, N.M.,  et al. (2017) Gold nanoparticles, radiations and the immune system: Current insights into the physical mechanisms and the biological interactions of this new alliance towards cancer therapy. Pharmacol Ther, 178, 1-17." . Such integrative discussions must be incorporated . 
  2. The authors need to support better the production of ROS for different NPs and energies of pulses which are currently in the MeV area. Very high. 
  3. The fact that the role of entering into the cell nucleus or as close as possible is completely disregarded opens logical questions about the use of these findings.

Answer:

We thank the reviewer for the valuable critics. Below we answer comment by comment, mentioning the manuscript' text modifications undertaken

* The line numbers are for the mode in which all corrections (track changes) are hidden.

  • We have revised the abstract and the introduction part, accordingly, to make them more precise.

Line 11: Gold nanoparticles are known to cause a radiosensitizing effect, which is a promising way to improve radiation therapy. However, the radiosensitization mechanism is not yet fully understood. It is currently assumed that gold nanoparticles can influence various physical, chemical, and biological processes. Pulse radiolysis is a powerful tool that can examine one of the proposed effects of gold nanoparticles, such as increased free radical production.

  • The radiosensitizing effect of AuNP has been shown for different sources of radiation: from X-ray to MeV energies. The applicability of 7MeV electron pulses as a source of ionizing radiation lies in the fact that high-energy electrons are low LET ionizing sources as gamma-rays. The energy transfer efficiency as a function of the energy for gamma and electrons in MeV ranges is almost flat; thus, there is almost no difference for 1MeV or 7MeV electron pulses or gamma quanta.

Concerning the different NPs, the reviewer did not mention what parameters must be considered. We can assume that the surface stabilization by bulk molecules could lead to a change in radiosensitizing properties of AuNP if the catalytic properties of AuNP are essential, as we proposed in our manuscript. The size effect will be only relevant in the context of surface catalyzed reactions by AuNP. In the context of physical dose enhancement, only the atomic gold concentration is essential, but not the size of an individual nanoparticle. Proposed surface-enhanced primary radicals' formation yields, in our understanding, have physical flaws since the formation yield of radicals is a fundamental property of the medium that cannot be altered significantly by the presence of surface since we are talking about the ionizing potential that is higher than 9eV for water.

As well we would like to stress out that in our work, we tested two types of AuNPs as well as silver nanoparticles that demonstrated similar results.

  • We do agree that our data is relevant mainly for homogeneous solutions. The text was modified accordingly, taking into account this comment.

Line 203: Of course, such a way of thinking is applicable only for homogenous solutions, and it is different for particles' aggregates usually formed in the cell.

Reviewer 3 Report

The authors start the paper by explaining the problems with current theories of how AuNPs act as X-ray radiosensitisers. However, that is not what is being tested in this experiment. Electron pulses were used, not X-ray pulses. The interaction of electrons with AuNPs is not expected to generate further secondary electrons (quite the opposite; it often results in electron quenching), while AuNP radiosensitisation with low-to-medium energy X-rays is presumed to occur because of the extra secondary electrons emitted from gold atoms. (I actually agree with the authors that gold nanoparticles probably act as catalysts for water radiolysis products, but that is not what is being tested here).

So, instead of quoting the literature on X-ray radiosensitisation (which is misleading in the context of this experiment), the authors should contextualise their experiment with respect to electron beam radiosensitisation papers. Because chloride ions will still be in solution after making the AuNPs, papers such as Hermannsdörfer et al., 2015 (https://doi.org/10.1039/C5CC06812F) would be appropriate.

Major point:

In order to show that AuNP supernatant is an appropriate control for the AuNPs, the authors need to quantify the density of AuNPs remaining in the supernatant, e.g. by TEM. They also need to check the concentration of any remaining gold salts in the supernatant, e.g. using ICP-MS.

Minor points:

Line 99: Please quote the evidence on the accuracy of solvated electron detection.

Line 157: Please reference that solvated electrons react with acetonedicarboxylic acid.

Line 164: Please reference that borate and borohydride anions scavenge OH radicals.

Line 167: …’the absence of any primary radical overproduction reported for similar AuNPs by indirect measurements’…  All the references quoted (5, 21,22) refer to X-ray radiosensitisation. Also, in the present experiments, the authors only look at solvated electrons – we don’t know what’s happening to radicals in this experiment.

Line 175: Differences in solution pH could be crucial. Please mention this information in the main text or Methods, not in a figure legend.

Line 178: …’and as a consequence, no OH radicals overproduction could be observed.’ The authors do not assay OH radicals in this manuscript. Therefore, this statement is misleading.

Line 184: …’interaction between radicals and particles could not occur in such conditions.’ It can occur, but will be a statistically rare event.

Line 190: …’then it must lead to a higher concentration of ROS.’ Not necessarily and not proven. In some cases, higher concentrations of radicals can lead to recombination and less ROS.

Line 191. Here the authors are discussing ionising radiation. However, this manuscript concerns electron beams. Where’s the data that electron beams lead to a greater production of secondary electrons around AuNPs?

Author Response

Reviewer 3

The authors start the paper by explaining the problems with current theories of how AuNPs act as X-ray radiosensitisers. However, that is not what is being tested in this experiment. Electron pulses were used, not X-ray pulses. The interaction of electrons with AuNPs is not expected to generate further secondary electrons (quite the opposite; it often results in electron quenching), while AuNP radiosensitisation with low-to-medium energy X-rays is presumed to occur because of the extra secondary electrons emitted from gold atoms. (I actually agree with the authors that gold nanoparticles probably act as catalysts for water radiolysis products, but that is not what is being tested here).

So, instead of quoting the literature on X-ray radiosensitisation (which is misleading in the context of this experiment), the authors should contextualise their experiment with respect to electron beam radiosensitisation papers. Because chloride ions will still be in solution after making the AuNPs, papers such as Hermannsdörfer et al., 2015 (https://doi.org/10.1039/C5CC06812F) would be appropriate.

Answer:

We are grateful for these comments and remarks which help us to improve our manuscript.

We tried to discuss X-ray radiosensitization in the introduction from a historical point of view, which in our opinion influenced the evolution of the understanding of AuNP radiosensitizing effect. We are citing mainly reviews on the radiosensitization considering X-ray and MeV sources of radiation. We have incorporated the proposed article in the manuscript.

Major point:

In order to show that AuNP supernatant is an appropriate control for the AuNPs, the authors need to quantify the density of AuNPs remaining in the supernatant, e.g. by TEM. They also need to check the concentration of any remaining gold salts in the supernatant, e.g. using ICP-MS.

We think that both gold salt reduction and the rest of AuNPs in its supernatant can be easier verified by UV-vis absorption technique. We have added an absorption spectrum on the supernatant to SI (Figure S1 A) and the following sentences to the method section

* The line numbers are for the mode in which all corrections (track changes) are hidden.

Line 92: Thus, a more relevant reference is a supernatant obtained by centrifugation of AuNPs suspension, eliminating gold nanoparticles to a large degree but leaving other chemical substances. We controlled by absorption measurement that at least 99% of AuNPs were removed from the supernatant solution (Figure S1).

Line 111: The full reduction of gold ions was verified by the total disappearance of the absorption at 220 nm.

Minor points:

Line 99: Please quote the evidence on the accuracy of solvated electron detection.?

The error of solvated electron detection in pico and nanosecond ranges equals to 1mO.D., while 5mO.D. for microsecond timescale. For pico and nanosecond timescale detection, the stroboscopic detection was used with an average 50 electron pulses for the one time-delay; whereas for microsecond timescale, Streak-camera detection with 400 electron pulses for the whole timescale was used. The accuracy of our setup is reasonable, taking in account the low frequency of the accelerator operation 5Hz and long optical passes for the probe light. As well in the cited works 16, 17 the general description of the setup is provided.

Line 157: Please reference that solvated electrons react with acetonedicarboxylic acid.

Line 164: ….because hydrated electrons react with acetonedicarboxylic acid (DCA) generated during the synthesis as a by-product of citrate oxidation [26]. Absorption at 260 nm in the supernatant verified the presence of DCA (Figure S1 B). The rate constant of the hydrated electron reaction with DCA is expected to be similar to the reaction with acetone (6.5×109 M-1s-1) [25].

Line 164: Please reference that borate and borohydride anions scavenge OH radicals.

Line 160: It is explained by the fact that anions BO33- and BH4- react with OH radical [25], suppressing OH radical-hydrated electron interaction, which increases the lifetime of e-aq.

Line 167: …' the absence of any primary radical overproduction reported for similar AuNPs by indirect measurements'…  All the references quoted (5, 21,22) refer to X-ray radiosensitisation. Also, in the present experiments, the authors only look at solvated electrons – we don't know what's happening to radicals in this experiment.

Ref. 5 is a review which includes works utilizing not only X-rays, but also 3 and 150 MeV protons, 6 MeV LINAC as well as gamma radiation source. Ref. 21 reports on •OH radicals' overproduction for both X-ray and γ-ray sources, and special properties of interface water are claimed to be responsible for such unexpected effect. Moreover, even in the case of X-rays the formation of •OH radicals was higher than predicted by physical dose enhancement, showing that the effect does not depend on the energy of ionizing radiation. We agree that ref. 23 is only about X-rays, so we will not link to this article to avoid any confusion.

Concerning the OH radical fate, we devoted to this subject a paragraph in the first version of the manuscript. It was stated that the decay of hydrated electron depends on the concentration of other radicals presented in the solution.

Line 175: Differences in solution pH could be crucial. Please mention this information in the main text or Methods, not in a figure legend.

Line 113: The solution of AuNPs prepared by borohydride reduction method had pH 8. The solution of AuNPs prepared by citrate reduction method had pH 4, which was adjusted by NaOH to 5.4.

However, the pH does not cause any change in the formation yields of the primary radicals at the first tens of picoseconds.

Line 178: …' and as a consequence, no OH radicals overproduction could be observed.' The authors do not assay OH radicals in this manuscript. Therefore, this statement is misleading.

Indeed, in this work, we do not perform a direct detection of OH radicals. However, as we discuss in the text, the concentration of solvated electrons just after the pulse corresponds to the same concentration of OH radicals produced from water ionization, and even if the "excitation" channel was hypothetically affected, any additional quantity of OH radicals would affect the solvated electron decay on the microsecond timescale. Therefore, we come up to the conclusion that OH overproduction is absent in the studied systems.

Line 184: …' interaction between radicals and particles could not occur in such conditions.' It can occur, but will be a statistically rare event.

We agree with the comment.

Line 186: thus the interaction between radicals and particles, as discussed elsewhere [29,30], it is a statistically rare event in our conditions.

Line 190: …' then it must lead to a higher concentration of ROS.' Not necessarily and not proven. In some cases, higher concentrations of radicals can lead to recombination and less ROS.

The absorbed dose can only be measured through the detection of produced radical species, and if it is claimed that the higher dose of radiation is absorbed, thus it automatically leads to a higher concentration of radical species for a particular source of radiation, having specific linear energy transfer (LET). The quantity of radicals at a given time depends on the dose rate and LET of the source.

However, we changed the sentence to be more precise.

Line 190: If AuNPs cause higher dose absorption, then it must lead to a higher primary radicals' formation, which could eventually result in a higher concentration of ROS later in time.

Line 191. Here the authors are discussing ionising radiation. However, this manuscript concerns electron beams. Where's the data that electron beams lead to a greater production of secondary electrons around AuNPs?

In the manuscript, we discuss both X-ray and γ-ray radiation as well as the electron beam. Of course, the electron beam is not commonly used in radiotherapy and in studies related to radiosensitization. However, the γ-rays and electrons of MeV energy represent a group of low LET radiation, comparable in action. In both cases, the spur formation is caused by secondary electrons interaction with the solvent molecules and not by primary high energetic particles, photons or electron, respectively.

Round 2

Reviewer 2 Report

In this revised version, the authors have partially considered comments and suggestions :

Especially these comments not necessarily minor :

The energies even in MeV region are very high. The increased cross section with gold of photons is in the keV region.

The authors must understand that when entering a cell the size is important. Also I have explained that nowadays the interaction with immune plays a role and especially when it comes to radiosensitivity. The limited use of one size impedes the application of their method. The measure of ROS comment is not really replied. 

Author Response

We thank the reviewer for helping us to make the manuscript more precise. We respond in detail to all the comments and made some changes to the text.

In this revised version, the authors have partially considered comments and suggestions :

Especially these comments not necessarily minor :

The energies even in MeV region are very high. The increased cross section with gold of photons is in the keV region.

To respond to this comment, we will use one example.

In this work (Gilles, M.; Brun, E.; Sicard-Roselli, C. Quantification of hydroxyl radicals and solvated electrons produced by irradiated gold nanoparticles suggests a crucial role of interfacial water. J. Colloid Interface Sci. 2018, 525, 31–38, doi:10.1016/j.jcis.2018.04.017.), gamma rays with 1.25 MeV energy were used. The reported result is the increased formation of 7-hydroxy coumarin in the presence of 32 nm AuNPs. It was interpreted as increased OH radicals and solvated electron production. The cross-section of gold and water is similar in this range of energy. Therefore, it was proposed that the higher radical production in the presence of AuNPs occurs due to the special properties of interfacial water.

It is well-known that gamma rays and 7 MeV electrons have the same LET (Linear energy transfer). Therefore, if overproduction of primary radicals was observed for gamma rays, it must be detectable for the pulse-radiolysis technique as used in our work.

Since we did not observe any changes in the production of primary water radicals, this makes doubt on the hypothesis published in many works (not only the one mentioned above) that AuNP can somehow lead to higher production of free radicals.

Line 207: Some of these works report a 4.5-fold enhancement of •OH radical production even for 1.25 MeV gamma rays, which have the same linear energy transfer as 7 MeV electrons used in our work [15].

The authors must understand that when entering a cell the size is important. Also I have explained that nowadays the interaction with immune plays a role and especially when it comes to radiosensitivity. The limited use of one size impedes the application of their method.

We agree with the reviewer that the size of nanoparticles is essential for cell studies. However,  in our work, we deal with Physico-chemical processes only. Any biological processes are out of the scope.

Line 216: We would like to stress out that the results of this paper must be utilized with care for cellular systems, where local dose enhancement around the AuNP must be taken into account for radiosensitization effect.

The measure of ROS comment is not really replied.

The main ROS are •OH, H2O2 and O2•-. The absence of additional OH radical formation in the presence of AuNPs is discussed in the text. The dismutation of two OH radicals can form H2O2. Therefore, the formation of H2O2 is not expected too. Superoxide radicals can be formed in the reaction of solvated electrons and molecular oxygen. So, solvated electrons were directly detected in our work, and it was shown that the presence of 20 and 45 nm AuNPs (3 mM atomic concentration) does not affect solvated electron formation.

Reviewer 3 Report

This manuscript is now suitable for publication

Author Response

We kindly thank the reviewer for his/her time for reviewing our work and helping us to improve it.

on behalf of the authors,

Dr. DENISOV Sergey